# Soil Microbial Community Responses to Different Management Strategies in Almond Crop

**DOI:** 10.3390/jof9010095

**Published:** 2023-01-10

**Authors:** Miguel Camacho-Sanchez, Juan F. Herencia, Francisco T. Arroyo, Nieves Capote

**Affiliations:** Andalusian Institute of Agricultural and Fisheries Research and Training (IFAPA), Center Las Torres, Alcalá del Río, 41200 Seville, Spain

**Keywords:** *Prunus dulcis*, almond agroecosystem, sustainable management, metabarcoding, phytopathogenic fungi, organic farming

## Abstract

A comparative study of organic and conventional farming systems was conducted in almond orchards to determine the effect of management practices on their fungal and bacterial communities. Soils from two orchards under organic (OM) and conventional (CM), and nearby nonmanaged (NM) soil were analyzed and compared. Several biochemical and biological parameters were measured (soil pH, electrical conductivity, total nitrogen, organic material, total phosphorous, total DNA, and fungal and bacterial DNA copies). Massive parallel sequencing of regions from fungal ITS rRNA and bacterial 16 S genes was carried out to characterize their diversity in the soil. We report a larger abundance of bacteria and fungi in soils under OM, with a more balanced fungi:bacteria ratio, compared to bacteria-skewed proportions under CM and NM. The fungal phylum *Ascomycota* corresponded to around the 75% relative abundance in the soil, whereas for bacteria, the phyla *Proteobacteria*, *Acidobacteriota* and *Bacteroidota* integrated around 50% of their diversity. Alpha diversity was similar across practices, but beta diversity was highly clustered by soil management. Linear discriminant analysis effect size (LEfSE) identified bacterial and fungal taxa associated with each type of soil management. Analyses of fungal functional guilds revealed 3–4 times larger abundance of pathogenic fungi under CM compared to OM and NM treatments. Among them, the genus *Cylindrocarpon* was more abundant under CM, and *Fusarium* under OM.

## 1. Introduction

Almond cultivation has undergone a revolution in the last decade by the application of new management techniques such as controlled irrigation, fertilization, pruning systems, harvest mechanization, use of more productive cultivars, high-density cultivation, and the use of phytosanitary treatments to prevent pests and plant diseases [1]. Organic management is considered a good alternative to conventional systems due to the production of good quality and healthy fruits with a lower environmental impact [2,3]. Moreover, the high added value of organic almond fruits, together with the increasing demands by consumers for healthy environmental practices and food safety, are a stimulus for this production system [1,4].

Organic farming aims to minimizing the use of synthetic inputs such as chemical fertilizers, herbicides and pesticides, enhancing the use of natural resources to minimize negative environmental impacts [5]. The use of cover crops and the incorporation of organic fertilizers are common practices in organic systems aimed at improving soil quality and fertility [6,7]. The use of cover crops and manure is progressively reemerging in agriculture as a sustainable alternative to synthetic agrochemicals. Their use has been positively related to improved soil quality, regulation of nutrient cycling and enhancement of the suppressiveness of soil against several pathogens [8]. Cover crops can be grown in the soil to reduce the leaching of nutrients, avoid erosion, improve soil structure and suppress weeds, or can be incorporated into the soil as nutrients. Further, the application of manure supplemented with biocontrol microorganisms can improve resistance against plant pathogens [9].

Differences in microbial communities and physical–chemical characteristics of organically versus conventionally managed soils have been reported [10,11,12]. In addition, cover crops and soil amendments alter soil bacterial and fungal communities [13,14,15]. Several meta-analyses point to an increased microbial diversity or changes in community composition due to organic farming [16,17,18]. In addition, organic amendments may act as disease-suppressive soils, decreasing the incidence or severity of soil-derived plant diseases [18,19,20,21]. Interactions in the soil are complex, and practices related to tillage, fertilization and irrigation regimes alter the soil microbiota and their related functions [22,23]. The soil microbiome can derive benefits, such as those that improve plant growth (PGPR bacteria), biological control, stress resistance or nutrient cycling [24,25], but also disadvantageous such as phytopathogenic activity. The understanding of how different agricultural practices affect the microbial communities informs us towards more sustainable and/or productive crops. The aim of this work was to evaluate the impact of management practices (organic/conventional/no-management) of almond crops on the diversity of their fungal and bacterial communities. These were characterized via metabarcoding sequencing and their total abundances estimated via quantitative PCR (qPCR). We hypothesize that soils under organic management host a richer microbial community, with suppressive characteristics compared to conventional management.

## 2. Materials and Methods

### 2.1. Almond Orchards and Sample Collection

Soil samples were collected in April 2020 from experimental almond plantations established in January 2016 in the Andalusian Institute of Agricultural and Fisheries Research and Training (IFAPA), located in the middle valley of the Guadalquivir River (Seville, southwestern Spain, 37°30′48″ N; 5°57′46″ W) (Figure 1). The soil of the experimental plots were loam soils (43% sand, 26% silt, 31% clay), classified as Entisol group Xerofluvent subgroup Typic: soils formed from recently deposited material with a dry moisture regime (United States NRCS, 2014). We considered two plantations under different farming managements, organic (OM) and conventional (CM), and a non-managed meadow (NM) adjacent to the plantations (Table 1). The plantations were approximately 500 m apart from each other. Plots under OM and CM had drip irrigation water (4760 m^3^ ha^−1^ year^−1^), while the NM plot was rainfed.

Fertilization in the organic plot consisted of the annual application of composed cattle manure (2 kg m^−2^ year^−1^) and the use of green cover crops composed of a mixture legume: cereal (*Vicia sativa* L.: *Avena sativa* L.; 75%:25%) in 2017 and 2019, alternated with bean cover (*Vicia faba* L.) in 2016, 2018 and 2020. Cover crops were sowed each year after the first rains in autumn. They were mechanically cut and incorporated into the soil in the following March or April with an orchard tractor. In the CM plot, fertilization consisted of the application of an NPK complex fertilizer (15-15-15, 150 kg ha^−1^) at flowering each year, there were no cover crops, and tillage was performed once a year, when weeds were treated by applying herbicides. Pests and diseases in the OM and CM plots were treated according to the Regulation (EU) 2018/848 of the European Parliament on organic production and labelling of organic products [26] and with the regulations for integrated production in Andalusia [27], respectively. The non-managed plot consisted of a meadow adjacent to the orchard plots. In each plot, three composite samples were taken in between tree lines. Each composite sample consisted of four soil cores randomly taken with a soil sampler (T-shape handle with metal cylinder of ~4 cm diameter), from the top 20 cm of soil after excluding the first 5 cm. They were immediately frozen at −80 °C at the IFAPA facilities. Homogenized soil was divided into two subsamples, one for the physical–chemical analysis and the other for genetic analysis of the microbial community.

### 2.2. Analysis of Physical–Chemical Soil Characteristics

Electrical conductivity (EC) and pH of the soil were measured in a 1:2.5 soil/water extract, using a conductivity/pH meter (Hanna Instruments, HI5521). Total N concentration was determined by Kjeldahl digestion [28], and the percentage of organic matter by potassium dichromate oxidation using the Walkley and Black method, modified by Jackson [29]. Available P was measured using the Olsen method [30]. Differences of physical-chemical measurements between managements were evaluated with ANOVA, followed by Fisher’s Least Significant Difference (LSD) tests at α = 0.05. Statistics were carried in Statistix 9.1 (http://www.statistix.com/; accessed on 14 March 2021).

### 2.3. Extraction and Quantification of Soil DNA

Prior to nucleic acid extraction, the soils were thoroughly mixed and three 250-mg aliquots of each soil sample were selected for DNA extraction. DNA was isolated using the DNeasy PowerSoil DNA isolation Kit (Qiagen, Hilden, Germany), following the manufacturer’s instructions. An extraction blank to check for cross-contamination was included. DNA was quantified using the Qubit High Sensitivity dsDNA Assay (Thermo Fisher Scientific, Waltham, MA, USA). Total DNA yield was calculated relative to grams of soil used for DNA extraction.

### 2.4. Real-Time PCR Quantification of Bacteria and Fungi

Quantitative real-time PCR reactions were performed in triplicate for each soil sample in 96-well plates using a CFX Connect thermocycler (Bio-Rad, Waltham, MA, USA). Reactions were performed in a total volume of 25 μL and contained 1× SensiFAST SYBR No-ROX mix (Bioline, Waltham, MA, USA), 0.1 mg/mL bovine serum albumin (BSA, Waltham, MA, USA), 0.1 μM each fungal or bacterial primer, and 10–65 ng of DNA template (~5 μL of DNA extracts, 2–13 ng/μL). Amplification of fungal ITS2 rRNA region was performed with primers ITS3/KY02 (5′-GATGAAGAACGYAGYRAA-3′) [31] and ITS4 (5′-TCCTCCGCTTATTGATATGC-3′) [32] to obtain amplicons of around 300 bp. For amplification of 16 S rRNA bacterial genes, primers Eub338 (5′-ACTCCTACGGGAGGCAGCAG-3′) [33] and Eub518 (5′-ATTACCGCGGCTGCTGG-3′) [34] were used to render amplicons with an average length of 180 bp. Amplifications were carried out at 95 °C for 10 min, followed by 40 cycles of 95 °C for 20 s, 55 °C (for bacteria) or 47 °C (for fungi) for 20 s, and 72 °C for 20 s. Melting curves (55 to 95 °C) were included for each run to verify the specificity of the amplifications. For quantification, standard curves were developed using ten-fold serial dilutions (1 ng to 0.1 fg) of genomic DNA extracted from soil bacterial (*Mesorhizobium* sp.) and fungal (*Fusarium solani*) pure cultures. Standards curves were included in the qPCR runs in triplicate. Relative ITS and 16 S copy numbers were estimated with the CFX Connect software (Bio-Rad) and relativized to grams of soil in the DNA extraction. Differences of ITS or 16 S between managements was evaluated with ANOVA, followed by Fisher’s Least Significant Difference (LSD) tests at α = 0.05.

### 2.5. DNA Metabarcoding Library Preparation and Sequencing

For fungi, the complete fungal ITS2 region of around 300 bp was amplified using the primers ITS3/KY02 and ITS4. For bacteria, a fragment of the 16 S rRNA region of around 300 bp was amplified using the primers 515 F (5′-GTGYCAGCMGCCGCGGTAA-3′) [35] and 806 R (5′-GGACTACNVGGGTWTCTAAT-3′) [36]. These primers had attached tails complementary to Illumina adapters to their 5′ ends.

Locus specific PCRs were carried out in a final volume of 25 μL, containing 5–33 ng of template DNA, 0.5 μM of the primers, 6.5 μL of Supreme NZYTaq 2× Green Master Mix (NZYTech, Lisboa, Portugal), and ultrapure water up to 25 μL. The reaction mixture was incubated as follows: an initial denaturation at 95 °C for 5 min, followed by 25 cycles of 95 °C for 30 s, 46 °C for 45 s, 72 °C for 45 s, and a final extension step at 72 °C for 7 min. Oligonucleotide indices with unique barcodes were ligated in a second PCR with identical conditions but only 5 cycles and 60 °C as the annealing temperature. All PCRs included negative controls with water instead of DNA templates. The PCR products were run on a 2% agarose gel stained with GreenSafe (NZYTech), and imaged under UV light to verify the library size.

Libraries were purified using the Mag-Bind RxnPure Plus magnetic beads (Omega Biotek, Beijing, China). Then, they were pooled in equimolar amounts according to the quantification in the Qubit dsDNA HS Assay (Thermo Fisher Scientific). The pool was sequenced in the MiSeq PE300 (Illumina, Waltham, MA, USA) at the AllGenetics and Biology S.L facilities (La Coruña, Spain).

### 2.6. Processing of Sequencing Data

Paired-end raw forward (R1) and reverse (R2) FASTQ reads were demultiplexed by sample. Indices and sequencing primers were trimmed during the demultiplexing step, and the quality of the FASTQ files was checked with FastQC (bioinformatics.babraham.ac.uk/projects/fastqc/; accessed on 1 February 2022). Amplicon reads were processed using QIIME2 (release 2020.8) [37].

DADA2 [38], implemented in QIIME2, was used to trim PCR primers, filter low-quality reads, determine amplicon sequence variants (ASVs) using a denoising algorithm, merge paired reads, and remove chimeric sequences. An OTU-like table with the number of times that each ASV was observed in each sample was generated. The taxonomy was assigned to ASVs using the UNITE reference database [39] for fungi and a pre-trained classifier of the SILVA reference database [40] for bacteria, both using the feature-classifier *classify-sklearn* approach implemented in QIIME2 [41]. We excluded the following ASVs: singletons, ASVs occurring at a frequency below 0.01%, unassigned sequences, eukaryotic sequences from chloroplast and mitochondrial origin, and sequences assigned only at the kingdom level for fungi. A Biological Observation Matrix file was obtained and imported into R 4.2.1 [42] using the package *phyloseq* 1.24.2 [43] for further analysis and plotting.

### 2.7. Microbial Diversity: Alpha and Beta Diversity

To explore the most frequent taxa in the biological matrix, ASVs were agglomerated by taxonomic ranks to generate relative abundance plots and tables. Before comparing alpha diversity between treatments, the read count for each sample was rarefied to the same number of sequences: 67,200 for bacteria and 34,800 for fungi. Alpha diversity was calculated over the rarefied matrix by reporting Shannon entropy, and the number of observed ASVs using QIIME 2. Kruskal–Wallis tests were used to assess differences between management practices on the Shannon index.

Differences in community composition (i.e., Beta diversity) among soil types were evaluated by using the Bray–Curtis distance to generate a dissimilarity matrix. Multidimensional Scaling (MDS) was run on the dissimilarity matrix to visualize community differences using *phyloseq::ordinate* in R. Non-parametric permutational multivariate analyses of variance (PERMANOVA) with 999 random permutations was performed to quantify the effect of soil treatments in community composition using *vegan::adonis* in R [44].

A Venn diagram was performed to show the exclusive and shared fungal and bacterial genera between managements using the package *ggVennDiagram* [45].

### 2.8. Linear Discriminant Analysis Effect Size (LEfSe)

LEfSe algorithm [46] was used to identify statistical and biologically significant taxa, at genus level or above, associated with any of the three treatments: organic, conventional, and without management. Analyses were performed in the Galaxy server (http://huttenhower.sph.harvard.edu/galaxy/; accessed on 20 June 2022) [47]. LEfSe results were filtered with a custom script to remove unnamed taxa. The threshold for the logarithmic Linear Discriminant Analysis (LDA) score was set at 3.0 and the Wilcoxon *p*-value at 0.05. We represented significant LDA scores in bar plots.

### 2.9. Functional Analysis of the Fungal Communities

Fungal ASVs were classified into functional groups based on their taxonomic assignment using the software FUNGuild [48] and its community-annotated database. As a result, fungal functional groups were classified into “guilds” with three confidence ranks (possible, probable, and highly probable). The ASVs that ended up without functional assignment were classified as ‘unmatched’. Only guilds considered as ‘probable’ and ‘highly probable’ were considered. To facilitate the interpretation of the results, from the final 38 guilds detected, those containing the words “Saprotroph” and “Animal” were merged into “Saprotroph” and “Animal related”, respectively, ending up in a total of seven different guilds.

## 3. Results

The sequencing depth for ITS and 16 S libraries averaged 70,296 reads (standard deviation, *s.d*.: 10,033) and 103,831 (*s.d*.: 12,637), respectively (Appendix A). After the denoising and filtering steps, a total of 1158 fungal and 5878 bacterial ASVs were determined.

### 3.1. Physical–Chemical Soil Characteristics

Soil pH was slightly higher and electrical conductivity (EC) notably higher in NM soils than in CM and OM soils, which showed very similar pH and EC values between them. The percentage of organic material was higher in NM and OM soils than in CM soils. Nitrogen was similar across soils. Conversely, CM soils contained 2-fold the content of phosphorous than OM soils and 4-fold than NM soils (Table 2).

### 3.2. Quantification of Bacteria and Fungi by qPCR

Fungi were significantly more abundant in organically managed (OM) soils than in soils under conventional management (CM). In non-managed soils (NM) a lower presence of bacteria and fungi were detected compared with OM soils, although the differences were only significant in the case of fungi. Soils under organic management had a higher fungi:bacteria ratio than CM or NM soils (Figure 2).

### 3.3. Alpha Diversity

There was no evidence of soil management affecting alpha diversity. Rarefaction curves on ITS and 16 S depict diversity saturates after around 10,000 reads, confirming that sequencing depth was enough for measuring diversity (Appendix A). Rarefied features per sample varied between 153 and 307 ASVs for fungi and 1133 and 1998 for bacteria (Table 3). Statistics on Shannon entropy supported an homogenous diversity across samples: bacterial, *H*(2) σ_hannon_ = 3.8, *p* = 0.15; fungi, *H*(2) σ_hannon_ = 0.27, *p* = 0.88.

For fungi, *Ascomycota* (72%) and *Basidiomycota* (17%) accounted for about 90% of fungi detected. *Ascomycota* was the dominant fungal phyla, with mean percentages of 84, 73 and 59% in CM, OM and NM soils, respectively. *Basidiomycota* were more abundant in NM soils (28%) than in CM (10%) and OM (13%) soils. *Glomeromycota* (8%) had the highest frequency in NM soils and *Mortierellomycota* (13%) were most abundant in OM soils (Figure 3A). The five most abundant phyla in bacteria accounted for around 78% of the total 16S reads. *Proteobacteria* (26%) was the dominant bacterial phylum across soils, followed by *Acidobacteriota* (18%), *Bacteroidota* (12%), *Gemmatimonadota* (11%), and *Actinobacteriota* (11%) (Figure 3D). At the family level, *Cladosporiaceae* was the most abundant fungal family in CM (23%), *Nectriaceae* in OM (31%) and *Chaetomiaceae* in NM soils (13%) (Figure 3C). In bacteria, *Gemmatimonadaceae* was the most abundant in CM (8%) and NM (7%), whereas *Xanthomonadaceae* (7%) was the most abundant under OM (Figure 3F). No bacterial genus dominated in relative frequency above 5% in the community. *Fusarium* (30%) predominated in OM orchards, while *Cladosporium* (23%) and *Chaetomium* (9%) predominated in CM and NM soils, respectively (Appendix A).

A total of 59 fungal and 151 bacterial genera were shared among soils under different managements, whereas 41 fungal and 42 bacterial genera were exclusive to OM soil, 30 fungal and 49 bacterial genera were specific to CM soil, and 32 fungal and 22 bacterial genera were only detected in NM soil (Appendix A).

### 3.4. Beta Diversity

Soil samples under the same management regime had similar fungal and bacterial communities (Figure 4). However, significant shifts in community structure between management were observed: 59% (*p* = 0.0038) and 63% (*p* = 0.0025) of the variation in the PERMANOVA test for the fungal and bacterial community compositions, respectively (Figure 4). In the MDS analysis, dimensions 1 and 2 accumulated 38.5% and 21.0% of the variance for fungi, and 41.3% and 24.5% of the variance for bacteria. Samples under the same management clustered together (Figure 4).

### 3.5. Taxonomic Groups Associated to Management Practices

We detected multiple bacterial and fungal groups characterizing soils from each management after the LEfSe analysis. Among the top groups in soils under OM were fungi from *Sordariomycetes*, *Hypocreales, Fusarium* and *Nectriaceae*, and bacteria belonging to *Actinobacteria*, *Micrococcales* and *Flavobacteriales*. For NM soils, fungi from *Glomeromycota* and bacteria from *Acidobacteriota* stood out, and especially *Vicinamibacteria*. Lastly, for CM soils, fungi from *Capnodiales* and *Cladosporium*, and bacteria from *Gemmatimonadota*, *Firmicutes* and *Bacilli*, heavily characterized their communities (Figure 5).

Regarding target potential pathogenic fungi, *Cylindrocarpon* had a moderate average abundance of 6% of reads in soils under CM, while it was not present in OM and NM plots. *Macrophomina* was nearly absent across all the samples, whereas *Fusarium* reached high abundances, up to 30 % in OM soils, 9% in CM and 4% in NM (Figure 6).

### 3.6. Functional Analyses of Fungi Guilds

Saprotrophs were the most abundant functional groups overall: they made up to 72% of the relative abundance in soils under OM. Plant pathogens were around 4–8 times more frequent in CM (45%) than under NM (11%) or OM (6%). The third and fourth ranked functional groups included fungi with potential symbiotic activity, classified as “Arbuscular Mycorrhizal” and “Endophyte”. Arbuscular mycorrhizal taxa were in low frequency in CM (6%) and OM (0%), but a maximal representation under NM (34%) (Figure 7).

## 4. Discussion

The intensification of agriculture in recent decades has had adverse impacts on ecosystems and human health [49,50]. This work was carried out to characterize soil fungal and bacterial communities in almond crops under different managements: organic (animal manure plus green covers incorporated in the soil with tillage), conventional (inorganic fertilizers plus tillage) and no management.

Alpha diversity was not lower in soils under conventional practices compared to organic management. Both richness and Shannon entropy varied within a narrow range between samples across the different managements (Table 3). This was contrary to our expectations and a handful of studies [16,17,18]. However, more extensive metabarcoding studies contrasting organic versus conventional management point to them having similar levels of microbial diversity for both fungi and bacteria, the differences between management practices being reflected in the community structure [12,25]. We do report around 60% of the variation in the community composition due to management for both fungi and bacteria (Figure 4). These differences are probably related to tillage and use of inorganic versus organic fertilizers, and their effect on the physical structure of the soil, nutrient input, and mineralization processes. Organic matter is often higher in non-tillage soils, as tillage increases the exposure of organic matter that is physically protected in microaggregates to biodegradation [51,52]. Indeed, physical–chemical analysis of our samples revealed the lowest organic content under CM, together with the highest phosphorus, probably derived from inorganic fertilizers. Despite larger amounts of organic matter being incorporated in OM soils through the tillage of green cover plus animal manure, the largest values of organic matter were found in non-tillage NM soils. No tillage can also increase nutrient retention in sandy loam soils [53], and multiple studies also confirm a higher organic matter content in organic farming than in conventionally managed systems [54,55,56]. As we show, this can have an impact on the microbial communities, as they have been demonstrated to respond differently to organic and inorganic fertilization [25] and tillage management [57].

Microbial DNA copies and fungal:bacteria ratios were higher in OM soils than in CM or NM soils. This agrees with some studies that state that conventional management, through the use of chemical fertilizers, generally reduces the fungal to bacterial ratio compared with organic fertilization [25,58,59]. Organic material and arbuscular mycorrhiza were more abundant in NM soils. No tillage has been reported as more beneficial for the development of rhizosphere fungi when compared with conventional tillage [60]. In our assay, OM soils were subjected to tillage only for the incorporation of the green cover to the soil to increase its fertility. However, the use of the cover as no-till mulching could be a desirable practice in the organic management of almond cultivation, not only for the preservation of potentially beneficial rhizosphere microorganisms, the reduction of the erosion processes and the increase of soil organic matter content in unplowed soils [57,61], but also for the benefits of mulching: conservation of the soil moisture, control of soil erosion, suppression of weeds and removal of the residual effects of pesticides, fertilizers, and heavy metals [62].

We identified multiple groups of bacteria in OM soils that have been associated in the literature with suppressive soils: orders *Sphingobacteriales* and *Flavobacteriales*, and the genus *Flavobacterium* [18] (Figure 5). OM also had associated a greater abundance of fungi from *Hypocreales*, in which is placed the genus *Trichoderma*. This group has a wide variety of antagonistic proprieties against fungi [63]. We report the greatest pathogenic fungi under CM and minimal under OM (Figure 7). A special focus on targeting pathogenic fungi known to affect almond crops revealed that *Cylindrocarpon* was most abundant in CM soils, while *Fusarium* was most abundant in soils under OM (Figure 6). *Cylindrocarpon* species can cause important diseases to woody crops, including almonds [64], the soil being an important source of inoculum for *Cylindrocarpon* pathogens [65,66,67]. On the other hand, several *Fusarium* species have been reported pathogenic for almonds: *Fusarium euwallaceae*, a beetle-borne vascular pathogen, causes Fusarium dieback in almonds in California [68]. The species *F. acuminatum*, *F. avenaceum*, *F. brachygibbosum*, and *F. californicum* are the causal agents of canker disease in cold-storage bare-root propagated stone fruit and nut trees in California, with diverse potential sources of inoculum throughout the nursery management process [69,70]. However, many soil-borne *Fusarium* strains are nonpathogenic saprotrophs [71] or act as biological control agents [72]. These results are in line with the use of organic amendments to enhance the suppressiveness of soils to control soil-borne plant pathogens [20,73,74].

## 5. Conclusions

Overall, this work highlights the importance of a transition from conventional to organic farming as a sustainable approach in almond cultivation, especially given a climate change scenario in which an increase in the relative abundance of soil-borne pathogens is expected with warming on a global scale [75]. Transformation from conventional to organic systems can initially affect the yield of the organic almond crop, although the differences with the conventional system are reduced over time [76]. The high added value of organic almonds, together with the growing demand for environmentally friendly practices and food safety, are additional reasons to promote the implementation or transition to organic farming in almond cultivation.

## Figures and Tables

**Figure 1 jof-09-00095-f001:**
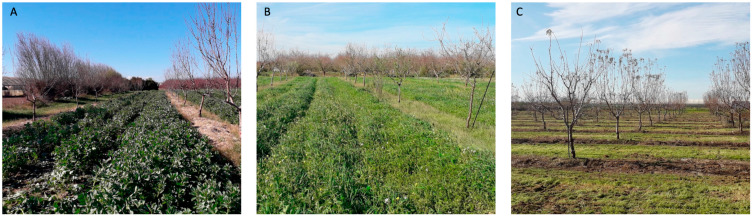
Almond plots under organic management with covers of *Vicia faba* (**A**) and *Vicia sativa:Avena sativa* (**B**), and under conventional management (**C**).

**Figure 2 jof-09-00095-f002:**
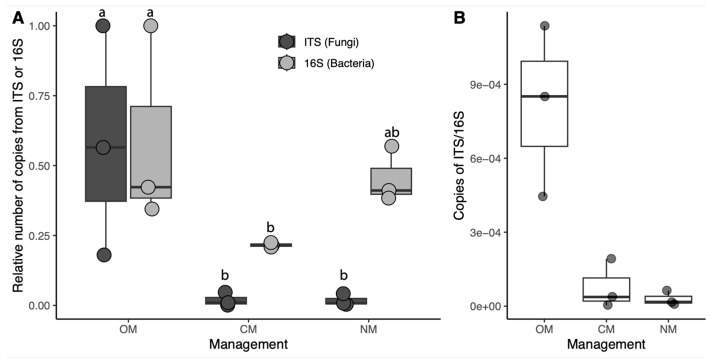
Fungal and bacterial abundances in the soil based on qPCR. (**A**) number of ITS or 16 S copies relative to the soil sample with largest abundance. Letters (a, b) indicate statistically different groups after LSD tests. (**B**) relative ITS/16 S copy ratio. OM, organic management; CM, conventional management; NM, no management.

**Figure 3 jof-09-00095-f003:**
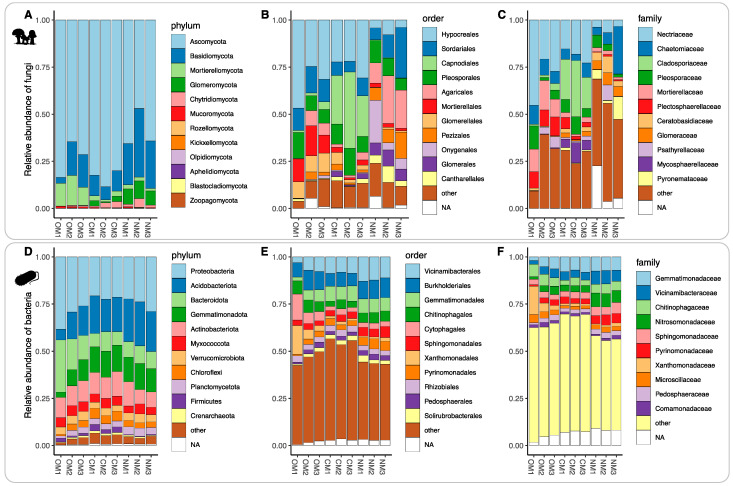
Alpha diversity. Relative abundance of major taxonomic groups for fungi (**A**–**C**) and bacteria (**D**–**F**) and across taxonomic ranks: phylum (**A**,**D**), order (**B**,**E**) and family (**C**,**F**). OM, organic management; CM, conventional management; NM, no management.

**Figure 4 jof-09-00095-f004:**
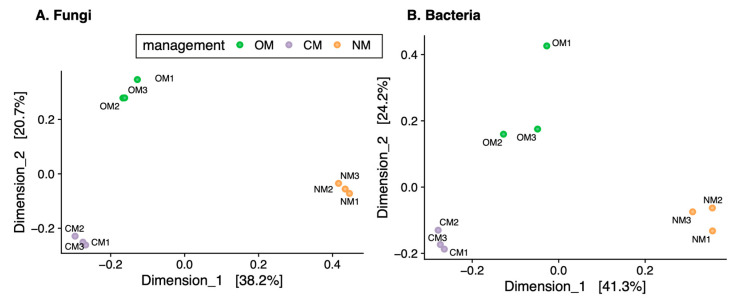
Beta diversity. Dimensions 1 and 2 from an MDS for fungi (**A**) and bacteria (**B**) using weighted Bray–Curtis distances. Percentage of explained variance for each dimension is depicted in brackets. OM, organic management; CM, conventional management; NM, no management.

**Figure 5 jof-09-00095-f005:**
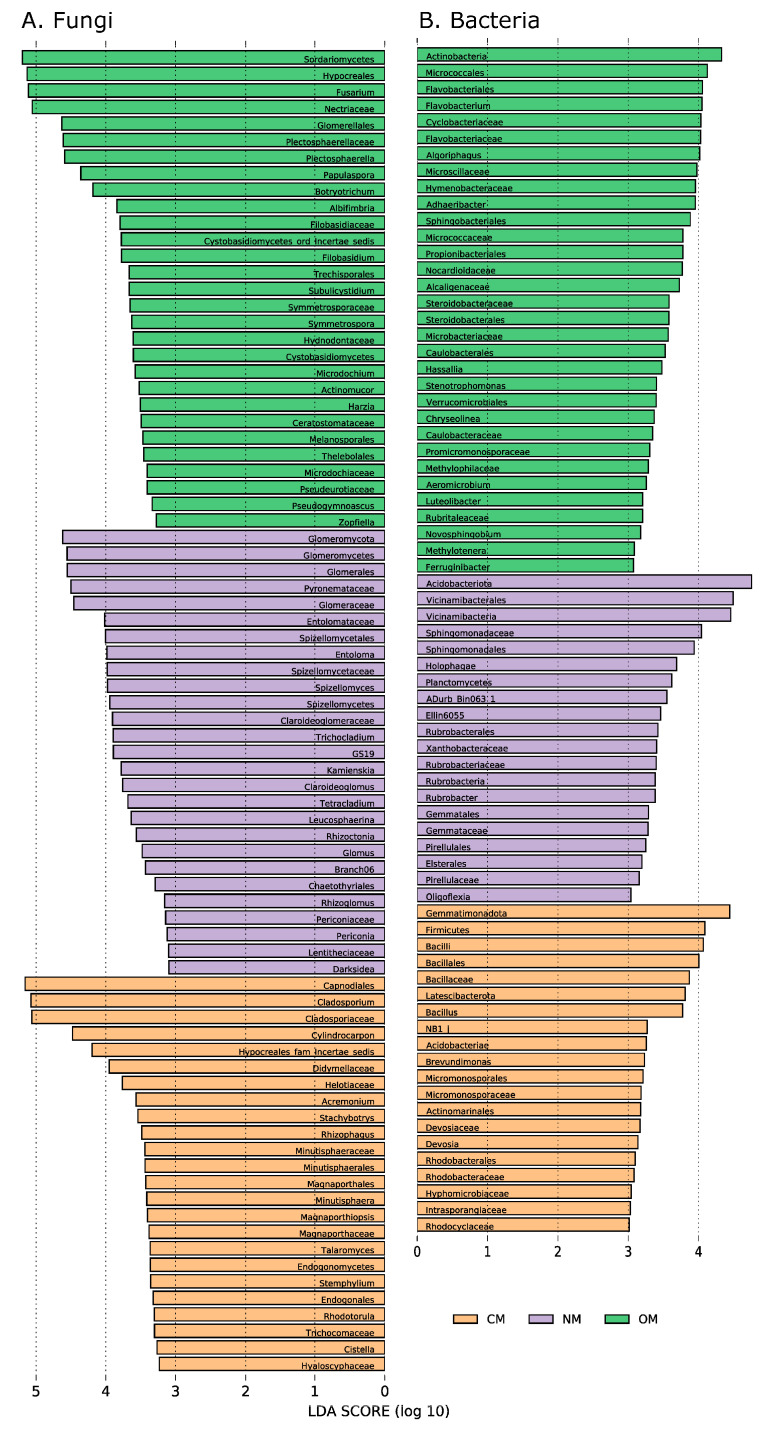
LEfSe analysis. The bars, ranked by LDA score, represent the taxa that most contributed to community differences between management practices for (**A**) fungi and (**B**) bacteria. Only taxa with an LDA score above 3.0 are depicted. OM, organically managed; CM, conventionally managed; NM, not managed soils.

**Figure 6 jof-09-00095-f006:**
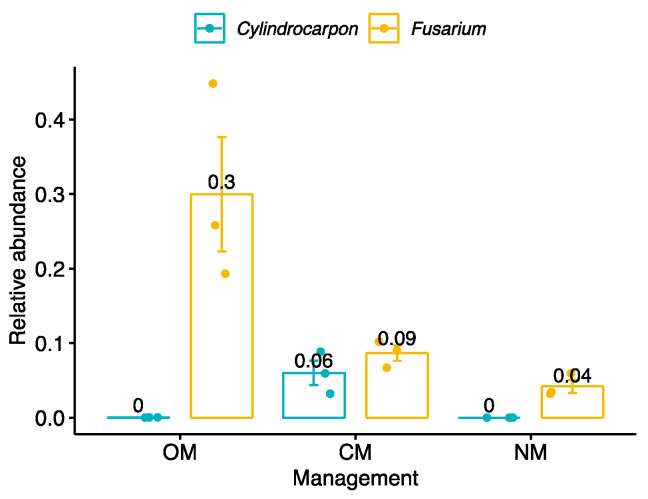
Relative abundance of the pathogenic fungi *Cylindrocarpon* and *Fusarium* across managements. OM, organic management; CM, conventional management; NM, no management.

**Figure 7 jof-09-00095-f007:**
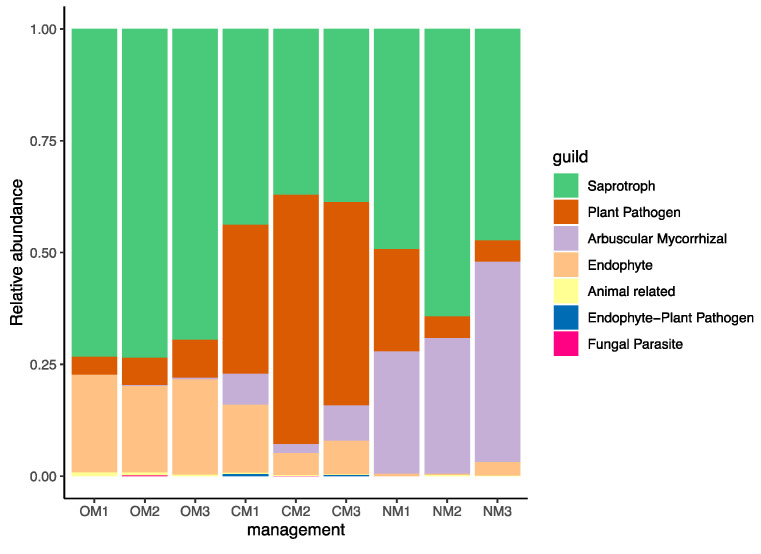
Relative abundance of fungal guilds according to their function after FUNGuild analysis. OM, organic management; CM, conventional management; NM, no management.

**Table 1 jof-09-00095-t001:** Management practices of the almond plots used in this study.

Management	Crop	Fertilization	Cover	Irrigation	Tillage
Organic	Almond	Animal manure ^1^	Legume/cereal ^3^	Drip irrigation ^4^	Yes
Conventional	Almond	Inorganic fertilization ^2^	No	Drip irrigation	Yes
None	Meadow	None	No	Rain fed irrigation	No

^1^ Composed beef manure (2 kg m^−2^ year^−1^); ^2^ N-P-K complex fertilizer (15-15-15, 150 kg·ha^−1^); ^3^
*Vicia sativa* L: *Avena sativa* L (75%: 25%) in 2017 and 2019, alternated with *Vicia faba* L. in 2018 and 2020; ^4^ Drip irrigation from March to October by using two pipelines with emitters of 2.3 L h^−1^.

**Table 2 jof-09-00095-t002:** Physical–chemical characteristics of almond crop soils under different managements. OM: organic management; CM: conventional management; NM: No management. Per column, different letters indicate significant differences according to LSD test (*p* = 0.05).

Sample Type	Soil pH	Electrical Conductivity (mS/cm)	Organic Material (%)	Nitrogen (%)	Phosphorous (ppm)
CM	7.65 ± 0.03 ^b^	140.75 ± 12.93 ^b^	1.06 ± 0.04 ^c^	0.07 ± 0.01 ^a^	33.98 ± 2.35 ^a^
OM	7.66 ± 0.02 ^b^	149.41 ± 14.14 ^b^	1.43 ± 0.14 ^b^	0.09 ± 0.01 ^a^	17.68 ± 1.50 ^b^
NM	8.28 ± 0.15 ^a^	354.93 ± 26.25 ^a^	2.13 ± 0.05 ^a^	0.08 ± 0.00 ^a^	7.98 ± 1.58 ^c^

**Table 3 jof-09-00095-t003:** Alpha diversity over rarefied read counts.

	Fungi		Bacteria	
Managenet ^1^	Observed	Shannon Entropy	Observed	Shannon Entropy
OM	153–307	4.0–6.0	1133–1998	8.6–10.3
CM	232–256	4.9–5.9	1862–1997	10.2–10.3
NM	223–270	5.1–6.7	1549–1609	9.5–9.8

^1^ OM, organic management; CM, conventional management; NM, no management.

## Data Availability

Raw FASTQ files and assembled sequences have been deposited in the NCBI under the BioProject PRJNA884175, with the accessions KFWT00000000 for fungal ITS2 data and KFWS00000000 for bacterial 16 S rRNA. Code and *phyloseq* objects with data were deposited in https://github.com/csmiguel/almond_organic (10.5281/zenodo.7515388, accessed on 14 March 2021).

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
