# Peer review of "Soil Microbial Community Responses to Different Management Strategies in Almond Crop"

_jof, 2023, doi:10.3390/jof9010095_

Round 1
Reviewer 1 Report
The reviewed manuscript points to the importance of the ecological type of almond cultivation, especially in terms of climate change. The authors indicate the possible benefits of an unconventional type of cultivation, not only in biological, but also in economic terms.
Lines 68 - 78: In this section 2.1 please, give information about type of soil
Line 92: More information about sampling please. Were the samples taken separately at different depths, and if so, from which layers. Whether composite samples were created by mixing sub-samples from a given soil layer.
Lines 330 - 341: This part of the discussion shows how important soil type information is when considering the impact of conventional and no tillage management system on soil biota.

Author Response
The reviewed manuscript points to the importance of the ecological type of almond cultivation, especially in terms of climate change. The authors indicate the possible benefits of an unconventional type of cultivation, not only in biological, but also in economic terms.
Lines 68 - 78: In this section 2.1 please, give information about type of soil.
ANSWER: We have given a detailed explanation for the classification of the soil: "The soil of the experimental plots were loam soils (43% sand, 26% silt, 31% clay) classified as Entisol group Xerofluvent subgroup Typic: soils formed from recently deposited material with a dry moisture regime (United States NRCS, 2014)"
Line 92: More information about sampling please. Were the samples taken separately at different depths, and if so, from which layers. Whether composite samples were created by mixing sub-samples from a given soil layer.
ANSWER: We provide a more detailed explation about the soil sampling: "In each plot, three composite samples were taken in between tree lines. Each composite sample consisted of four soil cores randomly taken with a soil sampler (T-shape handle with metal cylinder of ~4 cm diameter), from the top 20 cm of soil after excluding the first 5 cm. "
Lines 330 - 341: This part of the discussion shows how important soil type information is when considering the impact of conventional and no tillage management system on soil biota.
Thank you for pointing this out. We framed the discussion of our biological results taking into account the physical-chemical characteristics of the soil.
Reviewer 2 Report
The manuscript entitled “Soil microbial community responses to different management strategies in almond crop” is studying the effect of organic maters amendment to soil on fungal and bacterial communities in comparison with conventional farming systems of almond orchards. In general, the manuscript is suitable for publication in the Journal.
Specific comments
Line 37: to minimize
Line 40 : aimed at improving
Line 60: informs us
Line 65: host
Line 280: dimensions
Line 302: pathogens
Line 328: omit “a” in “ around a 60%”
Line 334: highest instead of largest
Line 336: were being
Line 345: reduces
Line 370: targeting
Line 373: crops

Author Response
Dear reviewer,
Thank you for your comments on the original version of the manuscript.
Please, find below my response:
Line 37: to minimize
“while minimizing” changed to “to minimize”.
Line 40 : aimed at improving
“to improve” changed to “aimed at improving”.
Line 60: informs us
“informs” changed to “informs us”.
Line 65: host
“hosting” changed to “host”.
Line 280: dimensions
“Dimensions” changed to “dimensions”.
Line 302: pathogens
“Pathogens” changed to “pathogens”.
Line 328: omit “a” in “ around a 60%”
“around a 60%” changed to “around 60%”.
Line 334: highest instead of largest
“highest” changed to “largest”.
Line 336: were being
“were” changed to “were being”.
Line 345: reduces
“reduce” changed to “reduces”.
Line 370: targeting
“target” changed to “targeting”.
Line 373: crops
“cops” changed to “crops”